# Carbon dioxide insufflation reduces the relapse of ulcerative colitis after colonoscopy: A randomized controlled trial

Yuriko Otake-Kasamoto[1], Shinichiro Shinzaki[1,2], Satoshi Hiyama[1], Taku Tashiro[1], Takahiro Amano[1], Mizuki Tani[1], Takeo Yoshihara[1], Takahiro Inoue[1], Shoichiro Kawai[1], Shunsuke Yoshii[1], Yoshiki Tsujii[1], Yoshito Hayashi[1], Hideki Iijima[1], Tetsuo Takehara[1]*

1 Department of Gastroenterology and Hepatology, Osaka University Graduate School of Medicine, Suita, Osaka, Japan, 2 Department of Gastroenterology, Faculty of Medicine, Hyogo Medical University, Nishinomiya, Hyogo, Japan

* takehara@gh.med.osaka-u.ac.jp

## Abstract

**Data Availability Statement:** All relevant data are within the manuscript and its Supporting Information files.

### Background and aim

Colonoscopy is necessary for diagnosing and surveilling patients with ulcerative colitis, though it may cause disease flares. Colonoscopy with carbon dioxide ($CO_2$) insufflation decreases abdominal discomfort; however, its effect on exacerbation incidence in ulcerative colitis remains unclear. Therefore, this study aimed to evaluate the colonoscopy effects using $CO_2$ insufflation in patients with ulcerative colitis.

### Methods

Overall, 96 remissive patients with ulcerative colitis (partial Mayo score $\leq 2$) who underwent total colonoscopy between March 2015 and December 2019 at Osaka University Hospital were enrolled and blindly randomized to the $CO_2$ (n = 45) and air (n = 51) insufflation group (UMIN-CTR, number: UMIN000018801). The post-procedural abdominal discomfort and the clinical relapse (partial Mayo score $\geq 3$) rate within 8 weeks were evaluated.

### Results

Baseline backgrounds did not differ between the groups. The mean abdominal fullness and pain scores were significantly lower in the $CO_2$ group than in the Air group immediately ($p = 0.0003$, $p = 0.0003$) and 30 min ($p < 0.0001$, $p < 0.0001$) after colonoscopy. While the overall clinical relapse rate remained unchanged between the groups, the clinical relapse rate at 8 weeks after colonoscopy was significantly lower in the $CO_2$ group than in the Air group in patients not in complete remission (Mayo endoscopic subscore $\geq 1$, $p = 0.049$; or partial Mayo score $\geq 1$, $p = 0.022$).

**Funding:** The authors received no specific funding for this work.

**Competing interests:** The authors have declared that no competing interests exist.

## Conclusions

CO$_2$ insufflation can reduce abdominal discomfort in remissive patients with ulcerative colitis and decrease clinical relapse at 8 weeks after colonoscopy for those not in complete remission.

## Introduction

Ulcerative colitis (UC) is an inflammatory bowel disease (IBD) that causes chronic inflammation of the large intestine. The prevalence of UC worldwide has increased over the last decade [1], especially in the East [2]. Endoscopy is performed to diagnose UC and monitor mucosal inflammation activity and treatment effects [3]. Additionally, because the long disease duration of UC is a risk for colitis-associated colorectal cancer [4, 5], regular surveillance with colonoscopy is necessary to detect precancerous dysplasia [6, 7]. Although colonoscopy is a very useful and indispensable tool for evaluating disease activity and surveilling cancer in patients with UC [3, 4], patients frequently experience disease flares after colonoscopy, sometimes leading to additional UC medications [8]. Colonoscopy in patients with active UC also carries a risk of intestinal perforation [9].

Previous randomized controlled trials (RCTs) reported that using carbon dioxide (CO$_2$) insufflation for colonoscopy is safe and associated with less post-procedural pain than air insufflation [10, 11]. Similarly, a prospective RCT showed that employing CO$_2$ insufflation rather than air insufflation caused less post-procedural abdominal pain and bloating at least 3 h after colonoscopy in patients with IBD [12]. Regarding the quality of endoscopy, CO$_2$ insufflation has been reported to increase the adenoma detection rate, a clinical indicator of colonoscopic performance [13]. However, CO$_2$ insufflation has not been widely used, and air insufflation remains the major method of insufflation for colonoscopy globally, except for some developed countries [12, 14], which is because CO$_2$ insufflation requires a CO$_2$ insufflator and supply of CO$_2$ gas, and is thus more expensive [15]. Furthermore, whether CO$_2$ insufflation reduces post-procedural pain and clinical relapse after colonoscopy in patients with UC remains unclear [15]. Therefore, this study aimed to compare the advantages of colonoscopy with CO$_2$ insufflation or air insufflation in patients with UC in remission regarding subjective symptoms after endoscopy and post-procedural clinical relapse.

## Methods

### Study design

This study was designed as a single-center, prospective, single-blinded, randomized controlled trial. The study was approved by the Institutional Research Ethics Board at Osaka University Hospital (No. 14372) and conducted in accordance with the Declaration of Helsinki. Written informed consent was obtained from all patients prior to their inclusion in the study. Moreover, this study was registered in the clinical registry (UMIN-CTR, number: UMIN000018801) (http://www.umin.ac.jp/ctr/). This study was designed and analyzed in compliance with the CONSORT (Consolidated Standards of Reporting Trials) guidelines (S1 Checklist).

### Protocol

Patients with UC in remission (partial Mayo score [PMS] ≤ 2) who were diagnosed according to the established diagnostic criteria [16] and underwent total colonoscopy using CF-H260AZI

or CF-HQ290ZI video colonoscope system manufactured by OLYMPUS (Tokyo, Japan) between March 2015 and December 2019 at Osaka University Hospital were enrolled in our study. Exclusion criteria included a history of colectomy, clinically active UC (PMS $\geq$ 3), undergoing total parenteral nutrition, a history of chronic obstructive pulmonary disease, sedated colonoscopy, under 15 years of age, in poor general condition, or judged by the doctors to be in a condition unsuitable for the study. The reason for excluding patients who underwent sedated colonoscopy was that it would be difficult to accurately evaluate changes in subjective symptoms immediately and 30 minutes after the colonoscopy in these patients. Furthermore, it was considered that the reduction of pain during the examination might affect the subsequent relapse rate. After scoring PMS and providing informed consent, patients were blindly randomized to the CO$_2$ insufflation (CO$_2$) or air insufflation (Air) groups. Because the button for CO$_2$ insufflation differs from that for air insufflation, the doctors knew the group the patients were in; however, the patients were unaware of the insufflation method used. After the standard bowel preparation using polyethylene glycol, the endoscopists performed the same procedures for all patients regardless of their assigned group. Stratified randomization was used to allocate the patients in a 1:1 ratio to the Air and CO$_2$ groups; and its adjustment factors included sex, age, and disease extent. Satoshi Hiyama was responsible for creating the randomization protocol. The data collection, management, and patient randomization were performed using a secure Research Electronic Data Capture (REDCap) database [17]. All the endoscopists who conducted the colonoscopies had more than 5 years of experience in performing a colonoscopy. Subjective symptoms immediately after colonoscopy and 30 min after colonoscopy were assessed by inquiry forms provided to each patient. Moreover, the PMS was calculated at 1 week and 8 weeks after colonoscopy. The window periods of the visit were + 1 week for Week 1 and +/- 2 weeks for Week 8. The authors had access to information that could identify individual participants during or after data collection in their usual medical care.

## Endpoints

We set the primary endpoint as the clinical relapse rate, defined as PMS $\geq$ 3, at 1 week after colonoscopy. Meanwhile, secondary endpoints were set as follows: clinical relapse rate at 8 weeks after colonoscopy, change in treatment after colonoscopy, changes in vital signs (blood pressure and oxygen saturation [SpO$_2$]) after colonoscopy, subjective symptoms (abdominal fullness and pain) immediately after colonoscopy and at 30 min after colonoscopy, which were scored from 0 to 5 using the visual analog scale, and cecum intubation rate/intubation time/total procedure time. Adverse events were defined as those events deemed by the physician to be harmful regardless of whether or not they were causally related to the protocol treatment, and the following were treated as serious adverse events: events resulting in death, life-threatening occurrence, and events that can result in permanent or marked disability or dysfunction.

## Sample size calculation

As a previous report indicated that 16% of patients with UC who underwent colonoscopy experienced symptom exacerbations and required additional UC treatment [8], we assumed an exacerbation rate of 16% in patients who received air insufflation and 5% in those who received CO$_2$ insufflation. Based on an $\alpha$ error of 0.05 and a power of 0.8, the number of cases required to detect a significant effect was estimated to be 119 in each group (total of 238). The number of patients with UC attending our hospital was approximately 250, and therefore we judged that an adequate case accumulation could be expected in 2 years. However,

accumulating the appropriate number of cases became difficult; owing to the increasing number of cases undergoing sedated endoscopic procedures, the steering committee discontinued the accumulation of cases after 4 years, twice the originally planned period for case accumulation. The total number of cases was 107 when patient recruitment ended.

## Statistical analysis

An intention-to-treat analysis was used to compare the outcomes between the two groups. Additionally, descriptive statistics were used to quantify the results. Continuous variables were presented by the randomized groups, by using the mean with standard deviation for normal distributed data and median with interquartile range (IQR) for non-normal distributed data. Differences among categorical values were analyzed using Pearson's $\chi^2$ test or Fisher's exact test, while differences between groups were analyzed using Mann–Whitney U test for non-parametric data. We used available-case analysis to deal with missing data. Measurements were analyzed with JMP software version 14.0.0 (SAS Institute, Cary, NC, USA). Differences were considered statistically significant when $p < 0.05$.

## Results

### Patients

Informed consent was obtained from 107 patients with UC, all of whom were randomized to either the $CO_2$ or Air group (Fig 1). The first patient was included in the study on September 3, 2015, and the last patient was randomized on September 20, 2019. Of the 107 patients, 6 underwent sedated endoscopy; 4 patients were in clinical remission when the informed consent was obtained, though their PMSs were $\geq 3$ at the time of the colonoscopy; 1 patient was unable to give accurate subjective symptoms owing to dementia. After excluding these 11 patients who did not fulfill the inclusion criteria, data for a total of 96 patients ($CO_2$ group; n = 45 and Air group; n = 51) were analyzed (Fig 1). All patients were performed the colonoscopy in the group they were originally assigned by randomization. The baseline characteristics, including sex, age, disease period, disease extent, clinical activity (PMS), endoscopic score (Mayo endoscopic subscore [MES]), and treatment, did not differ between the two groups (Table 1), and no adverse events occurred during the study period.

### Colonoscopy procedure

Because information regarding the colonoscopy procedure and subjective symptoms after the colonoscopy was missing in 5 patients, analyses of the colonoscopy procedure were only performed on the remaining 91 patients ($CO_2$ group; n = 43, Air group; n = 48). The cecum intubation rate was comparable between the $CO_2$ and Air groups (100% [43/43] vs. 97.9% [47/48]; $p = 0.34$). The $CO_2$ and Air groups did not differ significantly regarding the cecal intubation time (median [IQR]; 7 [8] min vs. 6 [6] min, respectively; $p = 0.11$), total procedure time (22 [16] min vs. 24 [13] min, respectively; $p = 0.66$), or in years of experience of the endoscopists performing the colonoscopies (7 [6] years vs. 7 [5] years, respectively; $p = 0.62$) (Table 2).

### Clinical relapse rate

The overall clinical relapse rates of this study participant (n = 96) were 8.3% (8/96) at week 1 and 7.3% (7/96) at week 8 (Fig 2A). Subsequently, we investigated the difference in the clinical relapse rate between the $CO_2$ and Air groups. Although the patients in the $CO_2$ group tended to have a lower relapse rate, the difference between the two groups was not significant at week 1 (4.4% vs. 11.8%; $p = 0.28$) or week 8 (2.2% vs. 11.8%; $p = 0.12$) (Fig 2B).

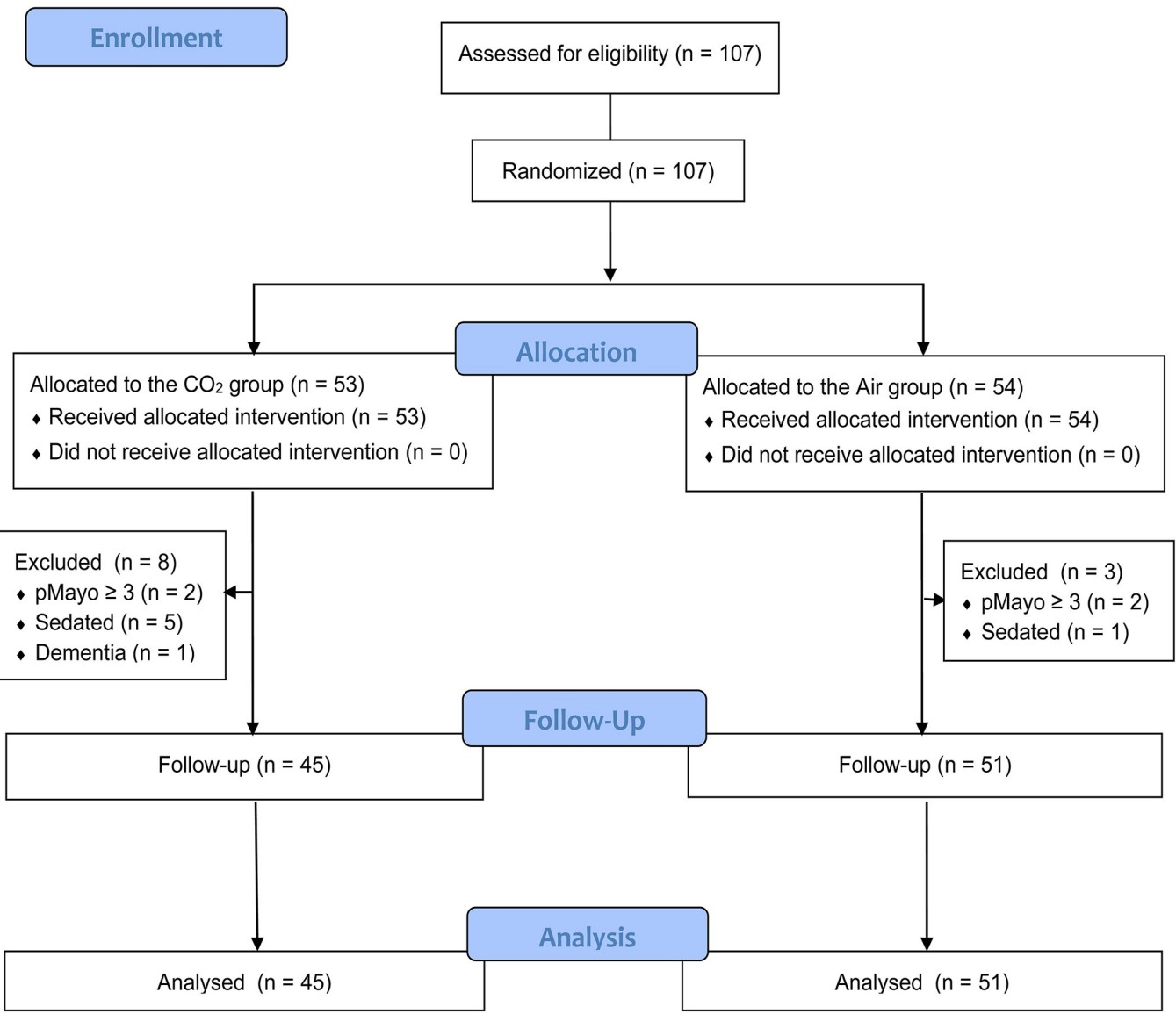

**Fig 1. Flow diagram of the study.** Overall, 107 patients with UC were recruited, all of whom were randomized into the CO$_2$ group or Air group. After excluding 11 patients, a total of 96 patients (CO$_2$ group; n = 45, Air group; n = 51) were analyzed.

Furthermore, to evaluate whether the severity of the mucosal inflammation identified by colonoscopy affected the post-procedural relapse rate, we divided patients according to the MES, which quantified mucosal disease activity from endoscopic findings. Among patients with MES 0, the clinical relapse rate did not differ between the two groups. Meanwhile, among patients with MES $\geq$ 1, the clinical relapse rate was significantly lower in the CO$_2$ group than in the Air group at week 8 (0% vs. 22.7%; $p$ = 0.049) (Fig 3A). In addition, to examine whether the clinical activity before colonoscopy affected the post-procedural relapse rate, we also divided patients according to the PMS. Among patients with PMS 0, the clinical relapse rate did not differ significantly between the two groups; however, among patients with PMS 1 or 2, the clinical relapse rate was significantly lower in the CO$_2$ group than in the Air group at week 8 (0% vs. 28.6%; $p$ = 0.022) (Fig 3B). Consequently, colonoscopy with air might be more harmful in patients who are not in complete remission at the time of the procedure.

**Table 1. Characteristics of the participants in the study.**

| | CO$_2$ group (n = 45) | Air group (n = 51) |
|---|---|---|
| Age, median (IQR) | 53 (33.5) | 46 (26) |
| Sex, male/female, n (%) | 27 (60)/18 (40) | 33 (65)/18 (35) |
| Disease period, median (IQR) (year) | 12 (13.5) | 11 (14) |
| Location of disease (Pancolitis/Left-sided/Proctitis), n | 25/13/7 | 26/13/12 |
| Type of clinical course (Relapse-remitting/chronic continuous/acute fulminating/first attack), n | 33/11/0/1 | 38/11/0/2 |
| Clinical activity (PMS 0/1/2) | 27/8/10 | 30/13/8 |
| Endoscopic activity (MES 0/1/2/3) | 24/15/5/1 | 29/14/8/0 |
| Medications | | |
| Mesalamine, n (%) | 40 (89) | 43 (84) |
| Systemic steroids, n (%) | 1 (2.2) | 1 (2.0) |
| Immunomodulators, n (%) | 8 (18) | 10 (20) |
| Anti-TNF-α (infliximab, adalimumab), n (%) | 4 (8.9) | 5 (9.8) |
| Tac/TOF/VED, n (%) | 0 (0) | 0 (0) |

IQR; interquartile range, MES; Mayo endoscopic subscore, PMS; partial Mayo score, Tac; tacrolimus, TOF; tofacitinib, VED; vedolizumab

## Subjective symptoms

Regarding subjective symptoms, we investigated the difference in post-procedural abdominal fullness and pain between the CO$_2$ and Air groups. The scores of abdominal fullness were significantly lower in the CO$_2$ group than in the Air group immediately after colonoscopy (1 vs. 2; $p = 0.0003$) and at 30 min after colonoscopy (0 vs. 2; $p < 0.0001$) (Fig 4A). Similarly, the abdominal pain scores were significantly lower in the CO$_2$ group than in the Air group immediately after colonoscopy (0 vs. 1; $p = 0.0003$) and at 30 min after colonoscopy (0 vs. 1; $p < 0.0001$) (Fig 4B). These results strongly suggest that, compared with air insufflation, CO$_2$ insufflation is associated with reduced abdominal discomfort after colonoscopy.

## Changes in vital signs

Systolic blood pressure (SBP), diastolic blood pressure (DBP), and peripheral SpO$_2$ were measured in all study participants at the baseline and end of the colonoscopy procedure. No severe changes in SBP, DBP, or SpO$_2$ were detected in either of the two groups, whereas the CO$_2$ group showed a significantly smaller change in the SpO$_2$ during the colonoscopy procedure than the Air group (0 vs. -1, $p = 0.0011$) (Table 3). These results indicate that neither CO$_2$ nor air insufflation induces severe changes in vital signs during colonoscopy.

**Table 2. Procedural characteristics.**

| | CO$_2$ group (n = 43) | Air group (n = 48) | p |
|---|---|---|---|
| Cecal intubation rate (%) | 43 (100) | 47 (97.9) | 0.34 [†] |
| Intubation time, median (IQR) (min) | 7 (8) | 6 (6) | 0.11 [‡] |
| Total time, median (IQR) (min) | 22 (16) | 24 (13) | 0.66 [‡] |
| Years of experience of the operators, median (IQR) (year) | 7 (6) | 7 (5) | 0.62 [‡] |

†; Pearson $\chi^2$ test

‡; Mann–Whitney U test, IQR; interquartile range

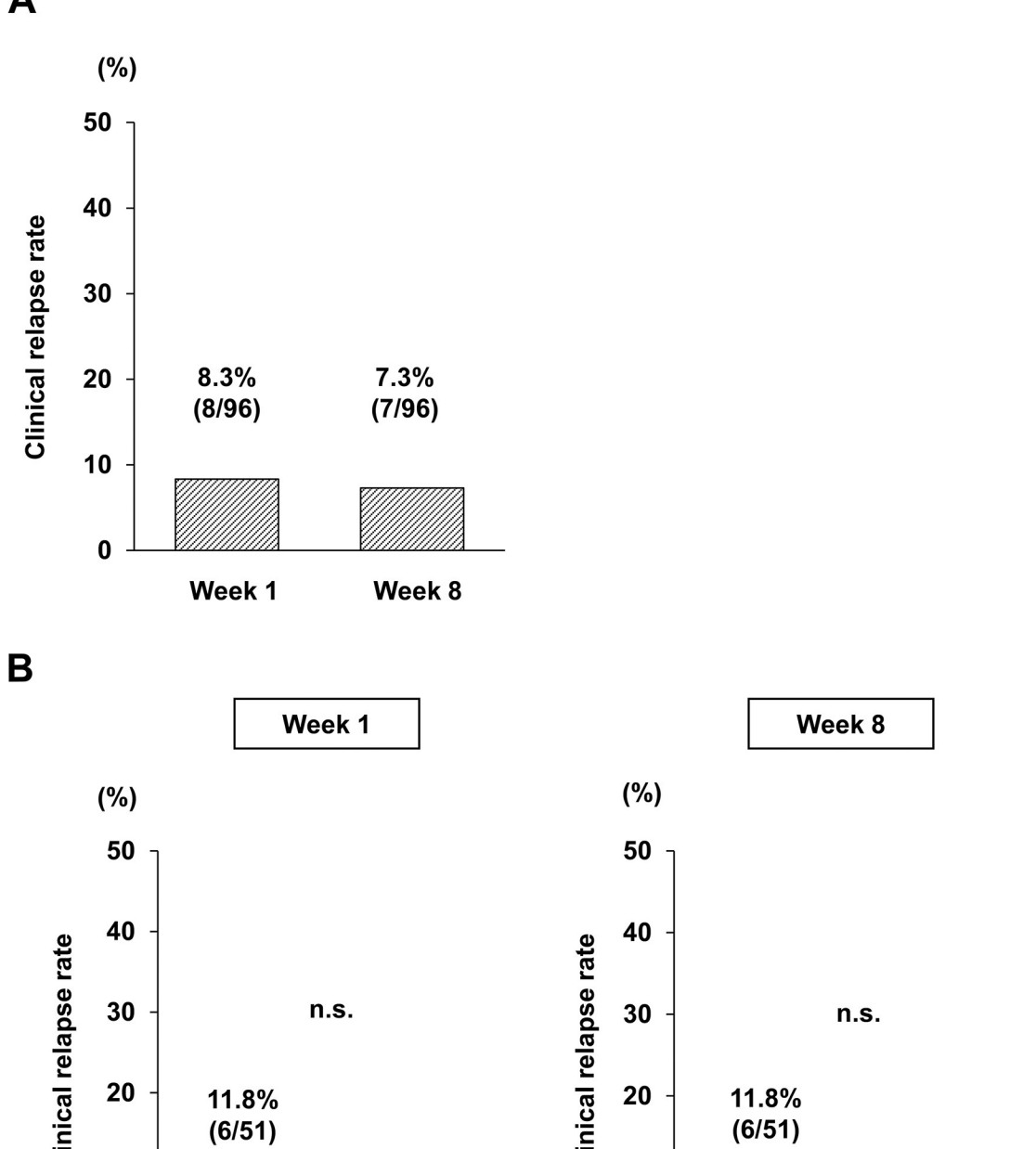

**Fig 2. Clinical relapse rate after colonoscopy.** (**A**) shows the overall clinical relapse rates of the study participants. (**B**) (left) indicates the clinical relapse rate at 1 week after colonoscopy. (right) The clinical relapse rate at 8 weeks after colonoscopy. Data were analyzed using Fisher's exact test.

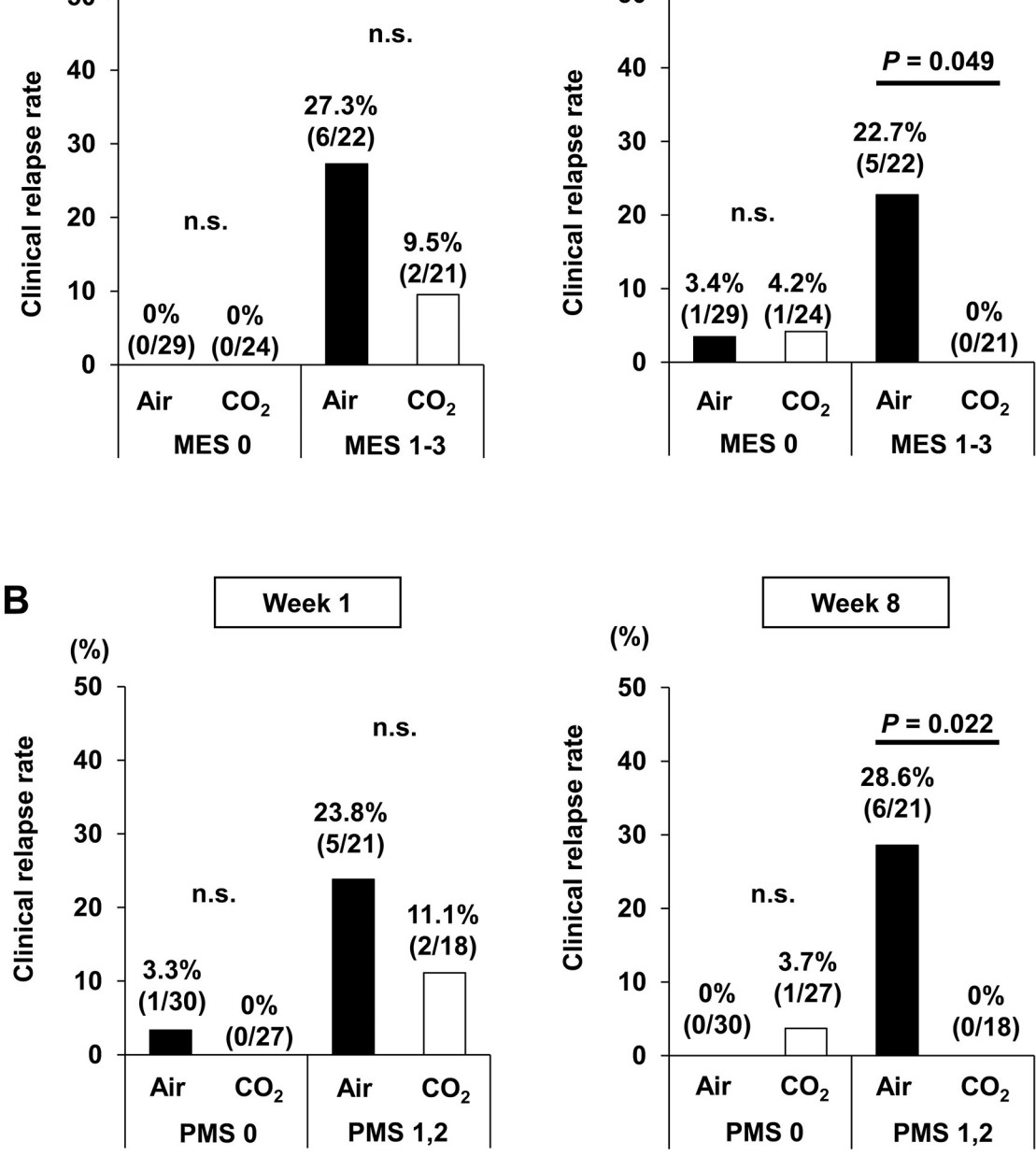

**Fig 3. Clinical relapse rate after colonoscopy adjusted by Mayo endoscopic subscore (MES) or partial Mayo score (PMS). (A)** indicates the clinical relapse rate at 1 and 8 weeks after colonoscopy based on the MES. Patients were divided into two groups, MES 0 and MES 1–3, and analyzed separately. n.s., not significant. Data were analyzed using Fisher's exact test. **(B)** shows the clinical relapse rate at 1 and 8 weeks after colonoscopy bad on PMS. Patients were divided into two groups, PMS = 0 or PMS = 1–2, and analyzed separately. n.s., not significant. Data were analyzed using Fisher's exact test.

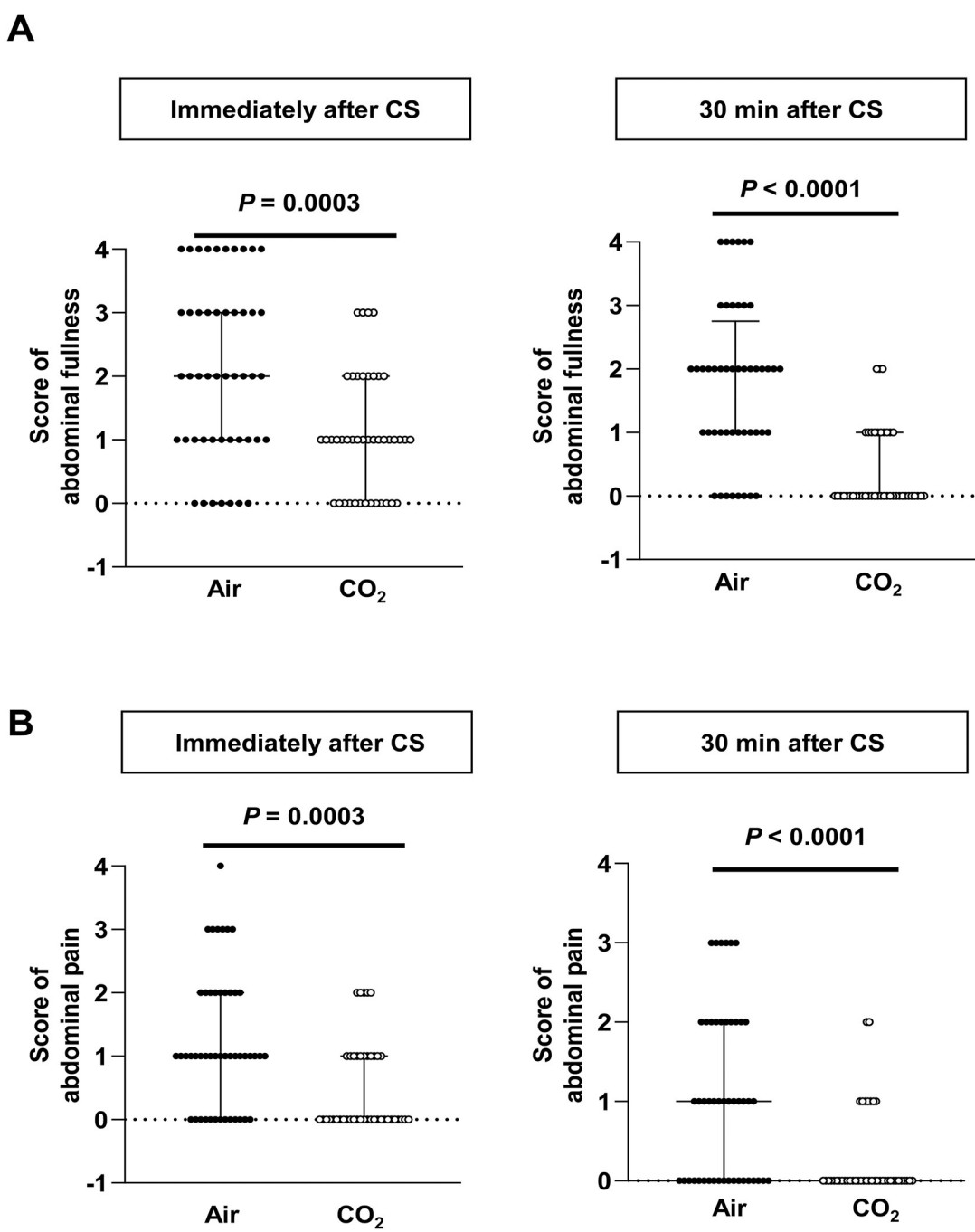

**Fig 4. Subjective symptoms after colonoscopy.** **(A)** shows the abdominal fullness scores immediately after (left) and at 30 min after (right) colonoscopy. Data were analyzed using the Mann–Whitney U test. The graphs show median with IQR. **(B)** indicates the abdominal pain score immediately after (left) and 30 min after (right) colonoscopy. Data were analyzed using the Mann–Whitney U test. The graphs show median with IQR.

## Changes in treatment due to clinical aggravation after colonoscopy

During the observation period, 4.4% of patients in the $CO_2$ group (2/45) and 7.8% of patients in the Air group (4/51) required more intensive treatment for UC after colonoscopy, and there were no significant differences between the two groups. The treatment changes included dose-

**Table 3. Changes in vital signs after colonoscopy.**

| | $CO_2$ group ($n$ = 43) | | | Air group ($n$ = 48) | | | $P$ |
|---|---|---|---|---|---|---|---|
| | Pre-CS | Post-CS | Change | Pre-CS | Post-CS | Change | |
| Systolic blood pressure, median (IQR) (mmHg) | 126 (26) | 118 (23) | -4 (18) | 123 (25.75) | 116.5 (27.25) | -4.5 (15) | 0.33 [†] |
| Diastolic blood pressure, median (IQR) (mmHg) | 74 (19) | 74 (22) | 0 (12) | 72 (17) | 75.5 (20) | +4 (14) | 0.15 [†] |
| $SpO_2$, median (IQR) (%) | 98 (2) | 98 (1) | 0 (1) | 98 (2) | 97.5 (2.75) | -1 (2) | 0.0011 [†] |

†; Mann–Whitney U test, IQR; interquartile range, CS; colonoscopy

escalation or types of 5-aminosalicylic acid (ASA) ($CO_2$ group, n = 2; Air group, n = 1), additional topical therapy of 5-ASA (Air group, n = 1), and additional topical therapy with prednisolone (Air group, n = 1) or hydrocortisone (Air group, n = 1).

## Discussion

This study evaluated discomfort following the colonoscopy procedure and the long-term disease relapse rate for approximately 8 weeks after colonoscopy in a single-blinded RCT. Subjective symptoms after colonoscopy were significantly different between the $CO_2$ and Air groups, consistent with findings from previous studies; however, the overall relapse rate did not differ between the $CO_2$ and Air groups for approximately 8 weeks after colonoscopy. Nevertheless, for patients not in complete remission (MES $\geq$ 1 or PMS $\geq$ 1), the clinical relapse was significantly lower in the $CO_2$ group at 8 weeks after colonoscopy.

Previous studies reported that colonoscopy performed with $CO_2$ insufflation was safe and reduced post-procedural pain [10, 18], which might be primarily due to the more rapid absorption and elimination of $CO_2$ from the colon, compared to colonoscopy performed using air insufflation [19]. An RCT on patients with IBD also reported that using $CO_2$ insufflation reduced subjective symptoms, including abdominal pain and bloating, at 1 and 3 h after colonoscopy [12]. However, the incidence of longer-term symptomatic worsening in patients with UC after colonoscopy with $CO_2$ insufflation has not yet been investigated. Our results suggest that air insufflation has more harmful effects in patients with subclinical intestinal inflammation, though it takes a comparatively long time until the effect manifests as a clinical relapse. Considering these findings, it may be more economical and preferable for patients and the environment to perform colonoscopy with air insufflation in patients with PMS 0 and $CO_2$ insufflation in patients with PMS $\geq$ 1.

The reason for the different relapse rates between groups at week 8 but not at week 1 is unclear; nevertheless, changes in the intestinal microbiota of the patients may provide a partial explanation. A recent RCT that examined changes in gut microbiota in fecal samples collected after colonoscopy with $CO_2$ or air insufflation reported that the relative abundance of some *Bacteroides* members, which were all obligate anaerobes, was significantly higher in the $CO_2$ insufflation group than in the air insufflation group [20]. Furthermore, previous studies reported that a decrease in normal anaerobic bacteria, including *Bacteroides*, *Eubacterium*, and *Lactobacillus* species, is associated with the pathogenesis of mucosal inflammation in patients with IBD [21, 22]. These results suggested that air insufflation may reduce some important anaerobic bacteria that cannot live in the presence of oxygen in the colon and exacerbate the activity of UC through host-pathogen immunological interactions. This process probably takes at least a few weeks, which may explain why the clinical relapse rate in the Air group increased 8 weeks after colonoscopy.

In this study, one patient had an MES = 3 with PMS = 0 at the beginning of the study. The PMS is known to well correlate with the full Mayo score, including the MES, and is widely

used because it is noninvasive [23]. However, there are a certain number of cases in which the PMS deviates from the MES. In this study, we found that patients with a high MES were more likely to be aggravated by air insufflation. Accordingly, even if a patient's PMS is low and the colonoscopy was started using air insufflation, switching to CO$_2$ insufflation when high disease activity is identified during colonoscopy should be considered. Actually, among the other patients with MES $\leq$ 2, the relapse rate tended to increase in patients with higher MES (S1 Fig).

Furthermore, although blood pressure changes did not differ between the two groups, changes in the SpO$_2$ during the colonoscopy procedure were significantly smaller in the CO$_2$ group than in the Air group. A previous study reported that the increase in the abdominal circumference at 15 min was significantly smaller after a colonoscopy performed with CO$_2$ insufflation than after a colonoscopy conducted with air insufflation and that CO$_2$ insufflation reduced the abdominal distention after colonoscopy [24]. The lower change rate in SpO$_2$ in the CO$_2$ group observed in our study may be due to the slight decrease in abdominal pressure, leading to lower pressure in the lung.

This study has some limitations. First, this was a single-center study and not double-blinded. In addition, the number of participants was relatively small because of the increase in the number of cases undergoing sedated endoscopic procedures, making it difficult to recruit the required number of patients despite nearly doubling the recruitment period. If there had been a larger number of participants, there might have been a significant difference in the primary endpoint. Second, the exact amount of air or CO$_2$ delivered into or suctioned from the colon during colonoscopy could not be measured in this study. Third, we did not specify the type of endoscopes and endoscopists in the study. This was a result of randomizing the study to reduce the impact of these factors on its results; however, the physicians were not blinded, which could have impacted its results. Fourth, the participants of this study only included patients with UC in clinical remission. Finally, we could not eliminate the selection bias and information bias such as the white coat effect, although we tried to reduce this effect by giving the patients inquiry sheets after the colonoscopy and having the office staff collect them, instead of having this inquiry be directly done by the endoscopists. More studies are needed in the future that will include patients with active UC and a double-blind RCT in patients undergoing sedated procedures.

## Conclusions

In conclusion, CO$_2$ insufflation could reduce abdominal discomfort after colonoscopy in patients with UC in clinical remission, and may be associated with a decrease in clinical relapse at 8 weeks after colonoscopy in patients not in complete remission (PMS $\geq$ 1 or MES $\geq$ 1).

## Supporting information

**S1 Checklist. CONSORT 2010 checklist of information to include when reporting a randomised trial\*.**
(DOC)

**S1 Text. Protocol of the study.**
(DOCX)

**S1 Fig. Bar graphs showing the clinical relapse rate that was analyzed separately by the Mayo endoscopic subscore.**
(TIF)

## Acknowledgments

We would like to thank Editage (www.editage.com) for English language editing.

## Ethical considerations

The study was approved by the Institutional Research Ethics Board at Osaka University Hospital (No. 14372). The study was explained orally and in an attached patient information sheet, and consent was obtained in writing. The consent was obtained after stating in the explanation document that participation in the study is voluntary, that no disadvantages will be incurred if the patient does not agree to participate, and that the patient has the right to withdraw the consent obtained for the study at any time without any disadvantages. For minor patients, consent was obtained from the patients themselves and their parents or guardians. This clinical research was conducted in compliance with the Ethical Guidelines for Clinical Research (Ministry of Health, Labor, and Welfare) and the Declaration of Helsinki (World Medical Association). Adverse events were defined as events deemed by the physician to be harmful regardless of whether or not they are causally related to the protocol treatment and the following were treated as serious adverse events.

a. Those resulting in death

b. Life-threatening

c. Those that result in permanent or marked disability or dysfunction.

In the case of a serious adverse event, the head of the facility and the principal investigator were notified immediately. When handling data related to the study, due consideration will be given to protect the confidentiality of the subjects.

## Author Contributions

**Conceptualization:** Yuriko Otake-Kasamoto, Shinichiro Shinzaki, Hideki Iijima, Tetsuo Takehara.

**Data curation:** Yuriko Otake-Kasamoto, Shinichiro Shinzaki.

**Formal analysis:** Yuriko Otake-Kasamoto, Shinichiro Shinzaki.

**Investigation:** Yuriko Otake-Kasamoto, Shinichiro Shinzaki, Satoshi Hiyama, Taku Tashiro, Takahiro Amano, Mizuki Tani, Takeo Yoshihara, Takahiro Inoue, Shoichiro Kawai, Shunsuke Yoshii, Yoshiki Tsujii, Yoshito Hayashi, Hideki Iijima.

**Methodology:** Yuriko Otake-Kasamoto, Shinichiro Shinzaki, Hideki Iijima.

**Project administration:** Yuriko Otake-Kasamoto, Shinichiro Shinzaki.

**Software:** Yuriko Otake-Kasamoto.

**Supervision:** Tetsuo Takehara.

**Visualization:** Yuriko Otake-Kasamoto, Shinichiro Shinzaki.

**Writing – original draft:** Yuriko Otake-Kasamoto.

**Writing – review & editing:** Shinichiro Shinzaki, Hideki Iijima, Tetsuo Takehara.

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
