## [Decision Letter · Decision Letter 0]

30 May 2023

PONE-D-23-12603Carbon dioxide insufﬂation reduces the relapse of ulcerative colitis after colonoscopy: a randomized controlled trialPLOS ONE

Dear Dr. Takehara,

Thank you for submitting your manuscript to PLOS ONE. After careful consideration, we feel that it has merit but does not fully meet PLOS ONE’s publication criteria as it currently stands. Therefore, we invite you to submit a revised version of the manuscript that addresses the points raised during the review process.

We look forward to receiving your revised manuscript.

Kind regards,

Antonio Brillantino

Academic Editor

PLOS ONE

Additional Editor Comments:

It is an interesting study that needs to be carefully reviewed according to reviewers' comments.

Reviewers' comments:

Reviewer's Responses to Questions

**Comments to the Author**

1. Is the manuscript technically sound, and do the data support the conclusions?

Reviewer #1: Partly

Reviewer #2: Partly

Reviewer #3: Yes

2. Has the statistical analysis been performed appropriately and rigorously? 

Reviewer #1: No

Reviewer #2: Yes

Reviewer #3: Yes

3. Have the authors made all data underlying the findings in their manuscript fully available?

Reviewer #1: No

Reviewer #2: Yes

Reviewer #3: Yes

4. Is the manuscript presented in an intelligible fashion and written in standard English?

Reviewer #1: Yes

Reviewer #2: Yes

Reviewer #3: Yes

5. Review Comments to the Author

Reviewer #1: This is an interesting study comparing the advantages of colonoscopy with CO2 insufflation or air insufflation in patients with UC in remission regarding subjective symptoms after endoscopy and post-procedural clinical relapse,

Some points to consider:

1. Primary endpoint defined as PMS >=1 after colonoscopy, was there a window before or after this week for logistical reasons? i.e 7 days (+/-1) .

2. In line 151, state the number of cases at the point when the trial steering made the decision to stop recruitment?

3. In the statistical analysis section include as a first sentence or two, that descriptive analysis will be presented by randomised groups (i.e mean (SDs) for normally distributed variables and median(IQR) for non-normal data..

4. Define your populations of analyses for primary outcomes (i.e as randomisation regardless of what treatment was received -ITT, However looks like the analysis population is excludes those patients those lost to follow-up?

5. Mention how missing data will be handled in the stats section?

6. The authors state randomisation accounted for sex, age and disease extent (this is vague and should be more explicit - i.e stratification or minimisation?. Also state the allocation ratio and who was responsible for creating the randomisation schedule?

7. Safety outcomes were considered, i.e adverse events so include this in stats section, after the secondary outcomes.

8. Also mention that that CONSORT guidelines were implemented to report the RCT in the stats section.

9. With the outcome subjective response - did the authors consider a “white coat effect” bias, how was this handled?

10. In results section, state the recruitment period (i.e start date to date last patient was randomised)

11. Table 1 - this is an RCT and not recommended to test between groups, any differences observed will be due to chance. Recommend to remove p-values.

12. Table 2 and Table 3 - the Mann Whitney was performed, then mean (SD) should not be presented and instead median (IQR), otherwise this is misleading in terms what exactly is being compared.

Reviewer #2: This article examines the impact of endoscopy on symptoms and relapses after endoscopy for patients with ulcerative colitis (UC) in remission between those undergoing endoscopy under standard Air and the CO2.

1. This reviewer understood no difference in intubation time between the CO2 and the standard Air groups in this study. However, was no aspiration performed during the endoscopy procedure in this study? This reviewer is sure that there are cases where suction is applied during endoscopy to remove cleaning fluid and residual stools. Therefore, it would be difficult to accurately measure the amount of Air and CO2 delivered to individual patients and assess the impact, even if the procedure time is the same. This is the most disadvantageous point in this study

2. Sedated endoscopy reduces patient stress. Hence, this reviewer recommends excluding data from patients who underwent endoscopy under sedation.

3. Four patients in each group are receiving anti-TNF antibody therapies. Is this treatment a regular regimen? Also, was there any difference in the tendency to relapse after endoscopy in patients receiving these advanced therapies compared to patients in the 5-ASA treatment alone group?

Reviewer #3: Although CO2 insufflation is increasingly used during colonoscopy, this study is original and interesting. It is regrettable that the number of patients studied is too small. As a result, the primary objective of the study is negative, whereas with the appropriate number of patients it might have been positive. This point needs to be addressed more widely in the discussion.

Why only patients who underwent colonoscopy without sedation were included?

One patient had an MES = 3, which is surprising given that the patients were in clinical remission. This point needs to be addressed in the discussion.

Interestingly, only patients with active lesions at endoscopy showed clinical recurrence. Even if the number of patients in each group is small, it would be interesting to know the percentage of clinical recurrence at one and eight weeks according to the MES observed (0, 1, or 2) during colonoscopy.

The conclusion should be more cautious and use the conditional form.

6. PLOS authors have the option to publish the peer review history of their article (what does this mean?). If published, this will include your full peer review and any attached files.

Reviewer #1: No

Reviewer #2: No

Reviewer #3: No

---

## [Author Response · Author response to Decision Letter 0]

28 Jun 2023

Editor:

Thank you for submitting your manuscript to PLOS ONE. After careful consideration, we feel that it has merit but does not fully meet PLOS ONE’s publication criteria as it currently stands. Therefore, we invite you to submit a revised version of the manuscript that addresses the points raised during the review process.

Reply: We really appreciate the editor’s comments. According to the reviewer’s suggestions, we have carefully revised the manuscript. We think we responded to all the concerns raised by the reviewers.

Reply: We have modified the manuscript and file names to meet PLOS ONE's style requirements.

Additional Editor Comments:

It is an interesting study that needs to be carefully reviewed according to reviewers' comments.

Reply: Thank you for the editor’s comments. “Point-by-point responses” to the reviewers’ comments have been provided below. We have carefully reviewed the reviewers’ comments and revised the manuscript accordingly.

Reviewer #1: 

This is an interesting study comparing the advantages of colonoscopy with CO2 insufflation or air insufflation in patients with UC in remission regarding subjective symptoms after endoscopy and post-procedural clinical relapse,

Some points to consider:

Reply: Thank you for your review and thoughtful comments regarding our manuscript. Our responses to your comments are as follows:

1. Primary endpoint defined as PMS >=1 after colonoscopy, was there a window before or after this week for logistical reasons? i.e 7 days (+/-1) .

Reply: We appreciate the reviewer’s comments. The window periods of the visit were +1 week for Week 1 and +/- 2 weeks for Week 8. We have added the appropriate description in the Protocol section (page 9, lines 129–130), as follows:

“The window periods of the visit were + 1 week for Week 1 and +/- 2 weeks for Week 8.”

2. In line 151, state the number of cases at the point when the trial steering made the decision to stop recruitment?

Reply: Thank you for pointing this out. The total number of cases was 107 when patient recruitment ended. We have added an appropriate statement in the Sample size calculation section (page 10, lines 157–158), as follows:

“The total number of cases was 107 when patient recruitment ended.” 

3. In the statistical analysis section include as a first sentence or two, that descriptive analysis will be presented by randomised groups (i.e mean (SDs) for normally distributed variables and median(IQR) for non-normal data..

Reply: Thank you for the reviewer’s comments. As per the reviewer’s suggestion, we have added a statement that clarifies that we conducted descriptive analysis on two randomized groups and we used “mean with standard error” for parametric data and “median with interquartile range” for non-parametric data presenting continuous variables in the Statistical analysis section (pages 10-11, lines 162-165), as follows:

“Additionally, descriptive statistics were used to quantify the results. Continuous variables were presented by the randomized groups, by using the mean with standard error for normal distributed data and median with interquartile range (IQR) for non-normal distributed data.”

4. Define your populations of analyses for primary outcomes (i.e as randomisation regardless of what treatment was received -ITT, However looks like the analysis population is excludes those patients those lost to follow-up?

Reply: Thank you very much for your significant indication. Six patients (Air = 3, CO2 = 3) whose inquiry sheets, which gathered their subjective symptoms could not be collected were excluded as “lost to follow-up” from analysis for primary and secondary outcomes in the previous unrevised manuscript. Of those 6 patients, one patient was not able to accurately fill out any of the inquiry sheets due to dementia, which prevented us from collecting the essential data for this study including PMS. Therefore, this patient was determined to meet the exclusion criteria (“judged by the doctors to be in a condition unsuitable for the study”) and excluded from the population of the analysis. We have mentioned this in the Patients section, (page 12, lines 178-179), as follows:

“1 patient was unable to give accurate subjective symptoms owing to dementia.”

For the remaining 5 patients, only the inquiry sheets on subjective symptoms immediately after and 30 minutes after colonoscopy were missing. The relapse rate at Week 1 and Week 8 could be evaluated based on their answers on the inquiry sheets; therefore, we have judged that these 5 patients should not be excluded.

In the revised version, we have added the 5 patients to the analysis for the relapse rate, and only excluded them from the analysis regarding colonoscopy procedure and discomfort after colonoscopy. We have mentioned this in the Colonoscopy procedure section, (page 14, lines 196–198), as follow:

"Because information regarding the colonoscopy procedure and subjective symptoms after the colonoscopy was missing in 5 patients, analyses of the colonoscopy procedure were only performed on the remaining 91 patients (CO2 group; n = 43, Air group; n = 48).”

Accordingly, we have modified the flowchart in Fig 1 and amended the results of the analyses in the manuscript in Table 1 and the Patients, Clinical relapse rate, and Changes in treatment due to clinical aggravation after colonoscopy sections. 

5. Mention how missing data will be handled in the stats section?

Reply: Thank you for pointing this out. Patients who met the exclusion criteria were first excluded. Then, we used available-case analysis to deal with the missing data. We have added a statement regarding handling the missing data in the Statistical analysis section (page 11, lines 167–168), as follows:

“We used available-case analysis to deal with missing data.”

6. The authors state randomisation accounted for sex, age and disease extent (this is vague and should be more explicit - i.e stratification or minimisation?. Also state the allocation ratio and who was responsible for creating the randomisation schedule?

Reply: Thank you for the reviewer’s thoughtful comments. The randomization in this study was performed using stratified randomization, and the allocation ratio was 1:1 between the Air and CO2 groups. Satoshi Hiyama was responsible for creating the randomization protocol. We have added this information to the manuscript in the Protocol section (page 8, lines 120–123), as follows:

“Stratified randomization was used to allocate the patients in a 1:1 ratio to the Air and CO2　groups; and its adjustment factors included sex, age, and disease extent. Satoshi Hiyama was responsible for creating the randomization protocol.”

7. Safety outcomes were considered, i.e adverse events so include this in stats section, after the secondary outcomes.

Reply: We appreciate the reviewer’s comments. We defined adverse events as those events deemed by the physician to be harmful regardless of whether or not they were causally related to the protocol treatment, and the following were treated as serious adverse events: events resulting in death, life-threatening occurrence, and events that can result in permanent or marked disability or dysfunction. As per the reviewer’s suggestion, we have added an appropriate statement in the Endpoints section (page 9-10, lines 140–144), as follows:

“Adverse events were defined as those events deemed by the physician to be harmful regardless of whether or not they were causally related to the protocol treatment, and the following were treated as serious adverse events: events resulting in death, life-threatening occurrence, and events that can result in permanent or marked disability or dysfunction.”

8. Also mention that that CONSORT guidelines were implemented to report the RCT in the stats section.

Reply: Thank you for the reviewer’s comments. As the reviewer suggested, we had added a sentence mentioning that this study was designed and analyzed in compliance with the CONSORT guidelines in the Study design section (page 7, lines 98–100), as follows:

“This study was designed and analyzed in compliance with the CONSORT (Consolidated Standards of Reporting Trials) guidelines.”

9. With the outcome subjective response - did the authors consider a “white coat effect” bias, how was this handled?

Reply: We appreciate the reviewer’s concern. We have tried to eliminate the white coat bias effect to the best of our ability by giving the patients inquiry sheets, which asked about their subjective symptoms, to fill out after the colonoscopy, and having the office staff collect them, rather than having the colonoscopists directly make this inquiry themselves. However, we think that this bias was not completely eliminated. We have mentioned this point in the Discussion section (page 23, lines 346-349), as follows:

“Finally, we could not eliminate the selection bias and information bias such as the white coat effect, although we tried to reduce this effect by giving the patients inquiry sheets after the colonoscopy and having the office staff collect them, instead of having this inquiry be directly done by the endoscopists.”

10. In results section, state the recruitment period (i.e start date to date last patient was randomised)

Reply: Thank you for pointing this out. The first patient was included in the study on September 3, 2015, and the last patient was randomized on September 20, 2019. As per the reviewer’s suggestion, we have added a relevant statement in the Patients section (page 12, lines 174–175), as follows:

“The first patient was included in the study on September 3, 2015, and the last patient was randomized on September 20, 2019.”

11. Table 1 - this is an RCT and not recommended to test between groups, any differences observed will be due to chance. Recommend to remove p-values.

Reply: Thank you for the important suggestion. As per the reviewer’s suggestion, we removed p-values from Table 1.

12. Table 2 and Table 3 - the Mann Whitney was performed, then mean (SD) should not be presented and instead median (IQR), otherwise this is misleading in terms what exactly is being compared.

Reply: Thank you very much for your comments. As the reviewer mentioned, “mean with SD” was changed to “median with IQR” on Mann–Whitney U test for non-parametric data in Table 2, Table 3, and the error bars on graphs in Figure 4 were changed to show median with IQR accordingly.

Reviewer #2: 

This article examines the impact of endoscopy on symptoms and relapses after endoscopy for patients with ulcerative colitis (UC) in remission between those undergoing endoscopy under standard Air and the CO2.

Reply: Thank you very much for your review and valuable comments regarding our manuscript. Our responses to your comments are as follows:

1. This reviewer understood no difference in intubation time between the CO2 and the standard Air groups in this study. However, was no aspiration performed during the endoscopy procedure in this study? This reviewer is sure that there are cases where suction is applied during endoscopy to remove cleaning fluid and residual stools. Therefore, it would be difficult to accurately measure the amount of Air and CO2 delivered to individual patients and assess the impact, even if the procedure time is the same. This is the most disadvantageous point in this study

Reply: Thank you for the significant concerns raised by the reviewers. As the reviewer pointed out, fluids, residual stools, and gases were suctioned during endoscopy, and the exact amount of delivered gas could not be accurately detected by only using the procedure time as a reference. Although it is difficult to accurately measure the volume of both the fluids and gases aspirated, we agree with the reviewer that it is an important limitation of this study and we have added this limitation in the Discussion section (page 23, lines 340-342), as follows:

“Second, the exact amount of air or CO2 delivered into or suctioned from the colon during colonoscopy could not be measured in this study.”

2. Sedated endoscopy reduces patient stress. Hence, this reviewer recommends excluding data from patients who underwent endoscopy under sedation. 

Reply: We completely agree with your suggestion. Patients who received sedated colonoscopy met the exclusion criteria and were excluded from the study. Therefore, their data were not included in the present analysis as shown in Figure 1, and we have mentioned this in the Protocol section of the manuscript (page 8, lines 111-114), as follows:

“The reason for excluding patients who underwent sedated colonoscopy was that it would be difficult to accurately evaluate changes in subjective symptoms immediately and 30 minutes after the colonoscopy in these patients. Furthermore, it was considered that the reduction of pain during the examination might affect the subsequent relapse rate.”

3. Four patients in each group are receiving anti-TNF antibody therapies. Is this treatment a regular regimen? Also, was there any difference in the tendency to relapse after endoscopy in patients receiving these advanced therapies compared to patients in the 5-ASA treatment alone group?

Reply: We appreciate the reviewer’s comments. All patients on anti-TNF antibody therapies were receiving regular anti-TNF antibody drugs as a maintenance phase. As shown below, anti-TNF antibody therapies did not affect the clinical relapse rate at Week 1 (p = 0.56) or Week 8 (p = 0.13). 

The relapse rate at Week 8 may have had a tendency to be slightly higher in patients on anti-TNF therapies, and this may be because those patients generally have high disease activity and are at high risk of relapse.

The clinical relapse rate at 1 week and 8 weeks after colonoscopy with or without anti-TNF antibodies treatment. n.s., not significant. Data were analyzed using Fisher’s exact test.

Reviewer #3: 

Although CO2 insufflation is increasingly used during colonoscopy, this study is original and interesting. 

Reply: Thank you very much for your review and fruitful comments regarding our manuscript. Our responses to your comments are as follows:

It is regrettable that the number of patients studied is too small. As a result, the primary objective of the study is negative, whereas with the appropriate number of patients it might have been positive. This point needs to be addressed more widely in the discussion. 

Reply: We appreciate the reviewer’s comments. As per the reviewer’s suggestion, we have mentioned the possibility that we might have had a positive result for the primary endpoint if the number of participants was large enough in the Discussion section (page 23, lines 339-340), as follows:

“If there had been a larger number of participants, there might have been a significant difference in the primary endpoint.”

Why only patients who underwent colonoscopy without sedation were included?

Reply: The reviewer’s question is very important. We expected that that it would be difficult to accurately assess changes in subjective symptoms immediately and 30 minutes after the colonoscopy in patients who underwent sedated colonoscopy. Moreover, we were concerned that the reduction of pain during the examination might affect the subsequent relapse rate. Therefore, patients who underwent sedated colonoscopy were excluded from the present study. We have added this information in the Protocol section of the manuscript (page 8, lines 111-114), as follows:

“The reason for excluding patients who underwent sedated colonoscopy was that it would be difficult to accurately evaluate changes in subjective symptoms immediately and 30 min after the colonoscopy in these patients. Furthermore, it was considered that the reduction of pain during the examination might affect the subsequent relapse rate.”

One patient had an MES = 3, which is surprising given that the patients were in clinical remission. This point needs to be addressed in the discussion.

Reply: We appreciate the reviewer’s valuable comments. As the reviewer point out, there are a certain number of cases in which endoscopic findings and clinical symptoms diverge in practice, and we should pay attention to those patients that have high endoscopic activity without any clinical symptoms. We have added this point in the Discussion section (pages 21-22, lines 318-325), as follows:

‘” In this study, one patient had an MES = 3 with PMS = 0 at the beginning of the study. The PMS is known to well correlate with the full Mayo score, including the MES, and is widely used because it is noninvasive[23]. However, there are a certain number of cases in which the PMS deviates from the MES. In this study, we found that patients with a high MES were more likely to be aggravated by air insufflation. Accordingly, even if a patient’s PMS is low and the colonoscopy was started using air insufflation, switching to CO2 insufflation when high disease activity is identified during colonoscopy should be considered.”

Interestingly, only patients with active lesions at endoscopy showed clinical recurrence. Even if the number of patients in each group is small, it would be interesting to know the percentage of clinical recurrence at one and eight weeks according to the MES observed (0, 1, or 2) during colonoscopy.

Reply: Thank you for the reviewer’s valuable comments. As per the reviewer’s suggestion, we have analyzed the proportion of the relapses separately for MES 0, 1, and 2, and found that patients with MES 2 had a higher proportion of relapse both at Week1 and Week 8 than those with MES 0 or MES 1. We have added the graphs below as S3 Fig., and mentioned the relevant information in the Discussion section (page 22, lines 325-326), as follows:

“Actually, among the other patients with MES ≤ 2, the relapse rate tended to increase in patients with higher MES (S3 Fig).”

S3 Fig. The clinical relapse rate analyzed separately by Mayo endoscopic subscore. 

The conclusion should be more cautious and use the conditional form.

Reply: We appreciate the reviewer’s valuable comments. We have modified the language used in the conclusions section as per your suggestion. Additionally, we have added the detailed condition "PMS ≥ 1 or MES ≥ 1" after the phrase "not in complete remission" in the conclusions section (pages 23-24, lines 354-357), as follows:

“In conclusion, CO2 insufflation could reduce abdominal discomfort after colonoscopy in patients with UC in clinical remission, and may be associated with a decrease in clinical relapse at 8 weeks after colonoscopy in patients not in complete remission (PMS ≥ 1 or MES ≥ 1).”

Thank you for reviewing our resubmission. We look forward to hearing from you and would be happy to make further changes, if required.

---

## [Decision Letter · Decision Letter 1]

4 Aug 2023

Carbon dioxide insufﬂation reduces the relapse of ulcerative colitis after colonoscopy: a randomized controlled trial

PONE-D-23-12603R1

Dear Dr. Takehara,

We’re pleased to inform you that your manuscript has been judged scientifically suitable for publication and will be formally accepted for publication once it meets all outstanding technical requirements.

Kind regards,

Antonio Brillantino

Academic Editor

PLOS ONE

Additional Editor Comments (optional):

According with reviewers suggestions, the article, in its current form, may be accepted for publication

Reviewers' comments:

Reviewer's Responses to Questions

**Comments to the Author**

1. If the authors have adequately addressed your comments raised in a previous round of review and you feel that this manuscript is now acceptable for publication, you may indicate that here to bypass the “Comments to the Author” section, enter your conflict of interest statement in the “Confidential to Editor” section, and submit your "Accept" recommendation.

Reviewer #1: All comments have been addressed

Reviewer #3: All comments have been addressed

2. Is the manuscript technically sound, and do the data support the conclusions?

Reviewer #1: Yes

Reviewer #3: Yes

3. Has the statistical analysis been performed appropriately and rigorously? 

Reviewer #1: Yes

Reviewer #3: Yes

4. Have the authors made all data underlying the findings in their manuscript fully available?

Reviewer #1: Yes

Reviewer #3: Yes

5. Is the manuscript presented in an intelligible fashion and written in standard English?

Reviewer #1: Yes

Reviewer #3: Yes

6. Review Comments to the Author

Reviewer #1: (No Response)

Reviewer #3: (No Response)

7. PLOS authors have the option to publish the peer review history of their article (what does this mean?). If published, this will include your full peer review and any attached files.

Reviewer #1: No

Reviewer #3: No

---

## [Editor Report · Acceptance letter]

9 Aug 2023

PONE-D-23-12603R1 

Carbon dioxide insufﬂation reduces the relapse of ulcerative colitis after colonoscopy: a randomized controlled trial 

Dear Dr. Takehara:

I'm pleased to inform you that your manuscript has been deemed suitable for publication in PLOS ONE. Congratulations! Your manuscript is now with our production department. 

Kind regards, 

on behalf of

Dr Antonio Brillantino 

Academic Editor

PLOS ONE